# Monitoring the Impact of Large Transport Infrastructure on Land Use and Environment Using Deep Learning and Satellite Imagery

Marko Pavlovic [1,2,†], Slobodan Ilic [1,*,†], Nenad Antonic [2,†] and Dubravko Culibrk [1,3,†]

[1] Institute for Artificial Intelligence R&D, Fruskogorska 1, 21000 Novi Sad, Serbia; marko.pavlovic@ivi.ac.rs (M.P.); dubravko.culibrk@ivi.ac.rs or dubravko.culibrk@smartcloudfarming.com (D.C.)
[2] Cinteraction, Nikolajevska 2, 21000 Novi Sad, Serbia; nenad.antonic@cinteraction.com
[3] Smart Cloud Farming, Rosenthaler Str. 72a, 10119 Berlin, Germany
[*] Correspondence: slobodan.ilic@ivi.ac.rs
[†] These authors contributed equally to this work.

**Abstract:** Large-scale infrastructure, such as China–Europe Railway Express (CER-Express), which connects countries and regions across Asia and Europe, has a potentially profound effect on land use, as evidenced by changes in land cover along the railway. To ensure sustainable development of such infrastructure and appropriate land administration, effective ways to monitor and assess its impact need to be developed. Remote sensing based on publicly available satellite imagery represents an obvious choice. In the study presented here, we employ a state-of-the-art deep-learning-based approach to automatically detect different types of land cover based on multispectral Sentinel-2 imagery. We then use these data to conduct and present a study of the changes in land use in two geopolitically diverse regions of interest (in Serbia and China and with and without CER-Express infrastructure) for the period of the last three years. Our results show that the standard image-patch-based land cover classification approaches suffer a significant drop in performance in our target scenario in which each pixel needs to be assigned a cove class, but still, validate the applicability of the proposed approach as a remote sensing tool to support the sustainable development of large infrastructure. We discuss the technical limitations of the proposed approach in detail and potential ways in which it can be improved.

**Keywords:** land cover; land use; deep learning; transport infrastructure; environment; artificial intelligence

## 1. Introduction

Land use, transport infrastructure, and the environment have a complex relationship, well documented in the literature [1,2].

Land-use change is the main cause of biodiversity loss globally, and it can disrupt the ecosystem services that support human wellbeing, resulting in a decline in air and water quality, an increase in air and water temperatures, and an increased risk of flooding. At the same time, transport accounts for a large portion of greenhouse gas (GHG) emissions, exacerbating these effects, but can also have a profound impact on land use. Achieving sustainable development and United Nations' aim for secure land tenure will require integration of both land- and transport-related policies with effective land-use and natural-environment monitoring systems.

Different indicators and sources of data can be used to monitor the effect of transport infrastructure on land use [2,3] and the impact of human activities on the environment [4].

*Monitoring the Effects of Human Activities on the Natural Environment*

When it comes to environmental monitoring, the widespread use of ground-based monitoring methods, which are, in practice, characterized by the high costs of implemen-

tation, time consumptive, labor-intense activities, and limitations in the ability to cover large regions of interest, is currently in place [5]. Advancements in technology, however, provide an alternative way of monitoring the state of the environment and keeping track of the transformations of the landscape through the use of remote sensing technology [6,7].

There are various remote sensing platforms available, including terrestrial platforms (e.g., farming robots, mobile mapping data, stationary photography, terrestrial laser scanning), airplanes, Unmanned Aerial Vehicles (UAVs), and satellites that are used to obtain data in different environmental monitoring studies [5].

Of these, only satellite remote sensing technology can be scaled up to support worldwide environmental monitoring on a regular basis. To support such efforts, a number of satellites specialized in gathering Earth observation data have been launched in recent years. To further stimulate the development of remote sensing tools based on data from these missions, many support open access to data.

These satellite missions typically acquire optical imagery (in the usual channels of red, green, and blue), but the main feature is multispectral images gathered by instruments with bands tuned so that they have performance far superior to optical sensors in various environments and ecosystem sensing applications. For example, one of the thirteen bands of Sentinel 2 is Ultra blue designed for observation of coastal regions and aerosols.

While the relationship between transport infrastructure and land use is complex, especially when long-term effects of infrastructure are considered, an abstract theoretical model does exist, albeit designed primarily for urban environments [2]. First, proposed by Wegener and Fürst [8], it is shown in Figure 1, as adapted by Bertolini [9].

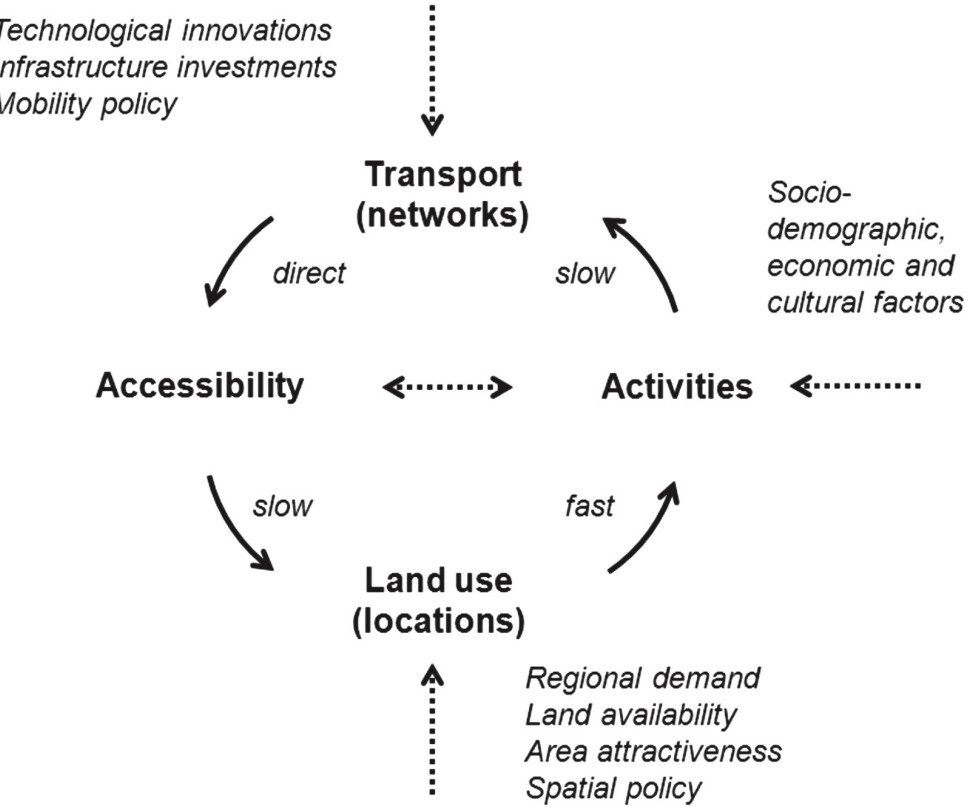

**Figure 1.** Transport and land use feedback cycle.

In simple terms, the development of transport infrastructure (be it rail or road) makes certain regions accessible, affecting land use. This in turn creates new human activities, which may then require new or different infrastructure to be put in place. However, numerous studies show that the effect of transport infrastructure on land use is mitigated by several factors.

Kasraian et al. [2] provide a good overview of studies on this topic, many of which deal with the significant effects of nascent rail infrastructure on land use during the historical periods in which this infrastructure was first being developed.

These days, railways span most inhabited parts of our planet and, although railway infrastructure is a dynamic system, there are few initiatives in the domain that promise to have a truly global impact. The exception to the rule is the development and expansion of the China Railway Express system that links China and Europe (CER) spanning the largest continent on earth. Developed under the auspices of the Belt and Road Initiative (BRI), which aims to cover a geographical area containing one-third of the world's wealth and 60% of its population [10], the project has the potential to become the most sustainable mode of long-range transport in Eurasia. Needless to say, this presents a unique opportunity to study the interaction of transport infrastructure, human society, and the environment, and CER is already drawing considerable attention among the research community [11,12]. Although CER is being actively developed, it already supports significant cargo transport. As of 5 November 2020, 10,180 trains were operating between China and Europe, for a total transport cargo volume of 9,277,000 TEU (Twenty-foot Equivalent Unit), up 54% from the previous year [12]. The railways that are currently in operation and the railways planned for development are shown in Figure 2.

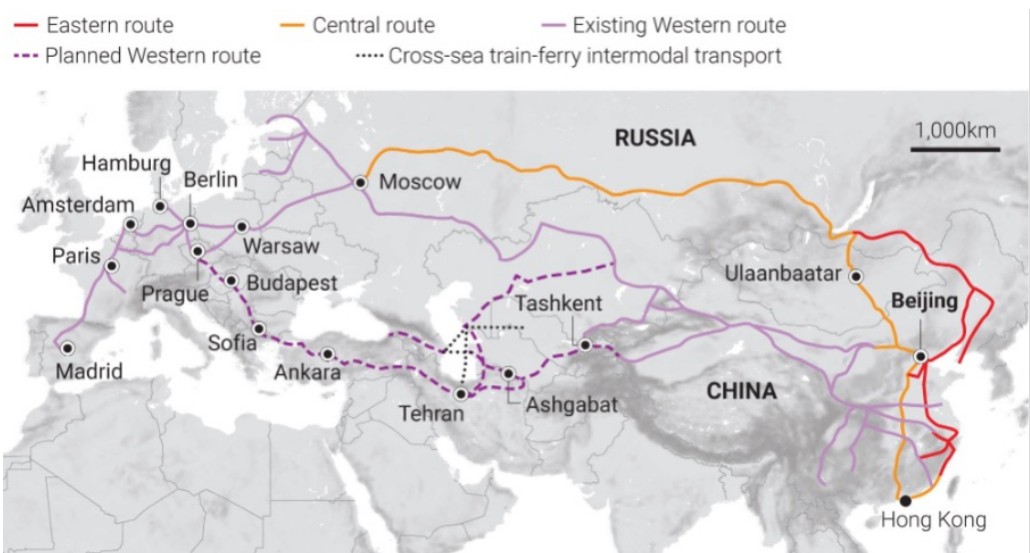

**Figure 2.** The planned routes of China Rail Express. Source: https://www.scmp.com/economy/global-economy/article/3169239/what-china-europe-railway-express-and-how-much-pressure-it, 27 April 2022.

Evaluating the impact of a project on such a massive scale requires a change in paradigm with respect to the research previously done. Deep learning approaches have the potential to provide the monitoring technology required to ensure adequate land administration and planning and support sustainable development along the CER route. Over the last decade, the area of artificial intelligence and machine learning, in particular, has been revolutionized by the advent of Deep Learning (DL) [13], paving the way for numerous applications of the technology in areas as diverse as computer vision, speech recognition, autonomous vehicles, etc.

Recently, deep learning concepts have started to be applied in remote sensing. Kattenborn et al. provide a review focused on vegetation sensing [5], while Hoeser and Kuenzer review the domain of DL object detection and image segmentation [14]. The latter concludes that, when it comes to deep learning, the entry barriers for earth observation researchers are high due to the dense and rapidly developing field mainly driven by advances in computer vision (CV). They try to tackle this problem by providing an overview of the evolution of DL with a focus on image segmentation and object detection

in convolutional neural networks (CNN). A more direct way of contributing to the solution of this problem is through the sharing of models and code for training and inference used in studies such as the one presented here. This is a common practice in the deep learning community and the importance of sharing is exacerbated by the need to accommodate the complexity of remote sensing data and the need to support different hardware platforms and frameworks. Thus, we open-source all our code and the model, which can be accessed at github (https://github.com/iai-rs/lulc-cnn-sentinel, 4 May 2022).

Land Cover (LC) monitoring forms a significant part of the environmental monitoring, environmental analysis, and applications and the development of accurate CNNs for land cover classification, trained on Earth observation data, represents a major part of the overall environmental protection effort [15]. However, such data are seldom used to evaluate the impact of transport infrastructure and we have not been able to identify a single published study where an automatic land cover detection approach was used for such a purpose; there is clear potential in using land cover as an indicator to study the interplay between transport, environment and land use. Coupled with the development of an adequately precise automatic LC detection solution, which we attempt to do in this study, such an approach could provide a continuous, cost-effective solution for global monitoring of this dynamic system. In our work, we adapt an existing open-source solution for land cover classification based on Sentinel 2 imagery and a well known VGG16 model to a new hardware platform (HUAWEI Ascend). We evaluate its performance on two regions along the existing and planned route of the China Express Railway, which is a major infrastructure project developed within the Chinese Belt and Road Initiative [10]. We provide our code and model for other researchers to further reduce the barriers faced by EO researchers when applying deep learning technology.

The rest of this paper is as follows: Section 2 provides an in-depth description of our approach. Our results are presented in Section 3 and discussed in Section 4. Finally, Section 5 holds our conclusions and some plans for future work.

## 2. Materials and Methods

To derive an effective tool for environmental monitoring based on land cover, we employ a CNN model trained to classify land cover based on Sentinel-2 imagery. This allows us to produce land cover data with the resolution of 10 m per pixel, i.e., at the resolution of the input images.

### 2.1. Sentinel-2 Data

Under the Copernicus Program, initiated in 2014, the European Space Agency launched a series of satellite missions, known as Sentinels. The Copernicus Program has three primary goals: production and dissemination of information in order to support EU global polices regarding environment and security, providing a platform for stockholders, providers and users for dialogue and collaboration and providing a legal, financial, organisational, and institutional framework for fluid functioning of European Space Agency satellite missions [16].

As our goal in this study is to detect land cover and land use changes, the appropriate data are those collected by the Sentinel-2 mission. The mission consists of two satellites: Sentinel-2A and Sentinel-2B. Both of the satellites are equipped with a Multi-Spectral Instrument (MSI), which measures reflected radiance from Earth surface in 13 spectral bands. Bands stretch from aerosols, over Near Infrared to the Short Wave Infra-Red (SWIR) spectral range with the spatial resolution ranging from 10 m to 60 m [17].

Table 1 lists all 13 Sentinel-2 bands and provides spatial resolution information for each band.

Sentinel-2A was launched in 2015 and followed by Sentinel-2B in 2017. The operational stage of the mission started in 2017, once the constellation was completed.

The key goals for the Sentinel-2 mission are to provide global and systematic collection of high-resolution multi-spectral imagery with a short revisiting period, to provide im-

proved continuity of multi-spectral imagery provided by the Satellite Pour l'Observation de la Terre series of satellites, and to provide information for the next generation of operational products such as land–cover maps and land–change–detection maps [18].

**Table 1.** Sentinel-2 spectral bands and their spatial resolution.

| Band Tag | Band Name | Spatial Resolution [m] |
| --- | --- | --- |
| B01 | Aerosols | 60 |
| B02 | Blue | 10 |
| B03 | Green | 10 |
| B04 | Red | 10 |
| B05 | Red edge 1 | 20 |
| B06 | Red edge 2 | 20 |
| B07 | Red edge 3 | 20 |
| B08 | NIR | 10 |
| B08a | Red edge 4 | 20 |
| B09 | Water vapor | 60 |
| B010 | Cirrus | 60 |
| B011 | SWIR 1 | 20 |
| B012 | SWIR 2 | 20 |

The design based on a pair of satellites enables the mission to cover the entire land surface of the Earth in approximately five days, as well as making it possible to collect remote sensing data for the same geographical point or area of interest every five days [19].

### 2.2. A Convolutional Neural Network for Land Cover Detection

Our system is based on the customization of the CNN-Sentinel open-source code, written and shared by Leitloff and Reise [20]. The original code supports the use of both VGG16 an DenseNet architectures [21], but we have opted for the more common VGG16 as the basis for our work [22].

As all convolutional neural networks, VGG16 incorporates consecutive convolution layers with a nonlinear activation function (in this case, the rectified linear activation function—ReLU), each followed by a pooling layer (max-pooling). These repeating convolutional blocks form the initial part of the network that is followed by fully connected layers, which perform the classification based on the output of previous layers, as shown in Figure 3.

The network contains a total of 13 convolutional layers and three fully connected layers.

The key aspects of the VGG architecture are the use of fitters with a small receptive field of 3 × 3 pixels, matching feature map size and number of filters in each convolutional layer of the same block and an increment in the number of the features extracted in the deeper layers, approximately doubling after a pooling layer [23].

The design of VGG16 is based on the famed Alexnet [24], but the VGG network utilizes a deeper architecture (doubling the number of layers), which enables it to recognize high level features and achieve higher classification accuracy [25].

Unfortunately, the 138 million parameters (network weights) in VGG11 require a labeled training data set, as well as the substantial computational power for the training procedure [23].

Our training and inference procedures were carried out on the Huawei platform Atlas 800, with two accelerator devices—Ascend 9000 Neural Processing Units (NPUs). Each NPU has a floating-point computing performance of 320 TFLOPS. Since the CNN-Sentinel code was intended for use on CPUs or NVIDIA GPUs—to use it on our platform, the code first had to be ported. Certain modifications needed to be done in order to properly initialize the NPU devices, calculate the loss function of the model, and satisfy additional constraints in terms of the supported versions of the TensorFlow framework. The resulting code can be accessed on github (https://github.com/iai-rs/lulc-cnn-sentinel, 4 May 2022).

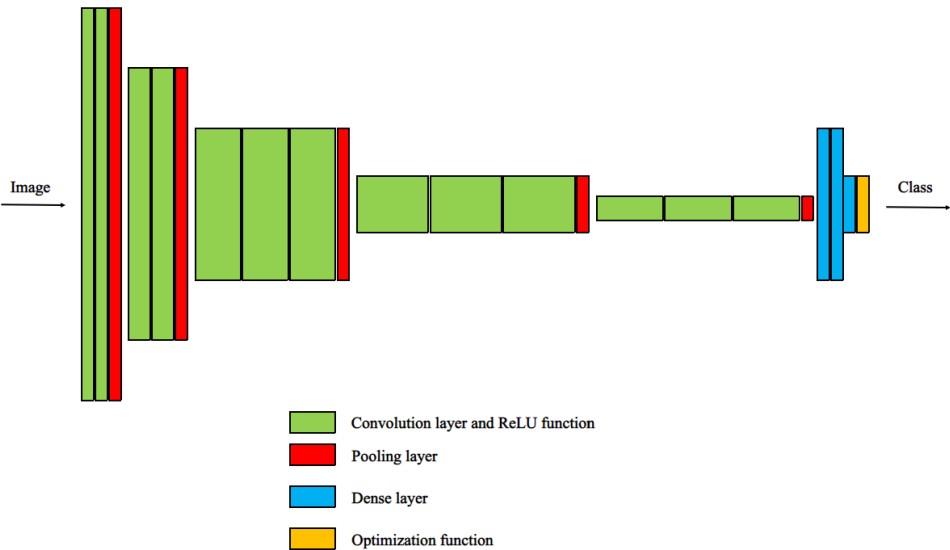

**Figure 3.** Architecture of the VGG network [5].

Our system is trained on the Eurosat data set [26]. Eurosat is a benchmark comprised of 27,000 images, with 2000–3000 images per land cover class, of which there are 10 in Eurosat (shown in Table 2).

**Table 2.** Definition of land cover classes for classification.

| LC Classes Coding and Color Representation | | | |
|---|---|---|---|
| **Class** | **Code** | **Hex Color** | **Color Patch** |
| Annual Crop | 0 | #ffe64d | |
| Forest | 1 | #00a600 | |
| Herbaceous Vegetation | 2 | #a6e64d | |
| Highway | 3 | #cc0000 | |
| Industrial | 4 | #cc4df2 | |
| Pasture | 5 | #e6e64d | |
| Permanent Crop | 6 | #ffe6a6 | |
| Residential | 7 | #a64d00 | |
| River | 8 | #00ccf2 | |
| Sea-Lake | 9 | #e6f2ff | |

The data itself represents 64 × 64-pixel patches of multi-spectral Sentinel-2 images, each assigned with a class label for the entire patch, but which can be interpreted as the classification of the central pixel of that patch. We used 70% of the data for training and left our 30% for validation.

Since the traditional VGG16 architecture is designed to work with standard RGB images, it had to be modified to incorporate 13-band multispectral data of Sentinel-2. The change to the architecture affects only the first convolutional layer as it needs to deal with higher-dimensional input. The rest of the network remains the same.

To be able to use the weights pre-trained on IMAGENET, rather than train the model from scratch, Leitloff and Reise devised a two-step training procedure. In phase I, all the weights of the pre-trained VGG16 model are copied, except those for the first convolutional layer and the fully connected layers, allowing the network to benefit from the feature extraction mechanisms trained on IMAGENET. The training in this phase, therefore, freezes the copied weights and trains only the first convolutional and the fully connected layers. Only in phase II, the fine-tuning of all the weights is done to improve performance. This allows us to reach the 97% classification accuracy for our model, which surpasses the benchmark accuracy of all RGB-based models tested in the original Eurosat paper [26],

where even the more complex architecture such as GoogleNet [27] and Resnet 50 [28] was employed. With the much more complex Resnet 50 model and a hand selection of three specific bands, the best result achieved by the authors was 98.57%, but we felt that such a small improvement in accuracy is not worth the additional complexity.

Using an Ascend 910 NPU, the training of our model is done in less than a day.

Once trained, the system can be used to generate false-color output images as the one shown in Figure 4.

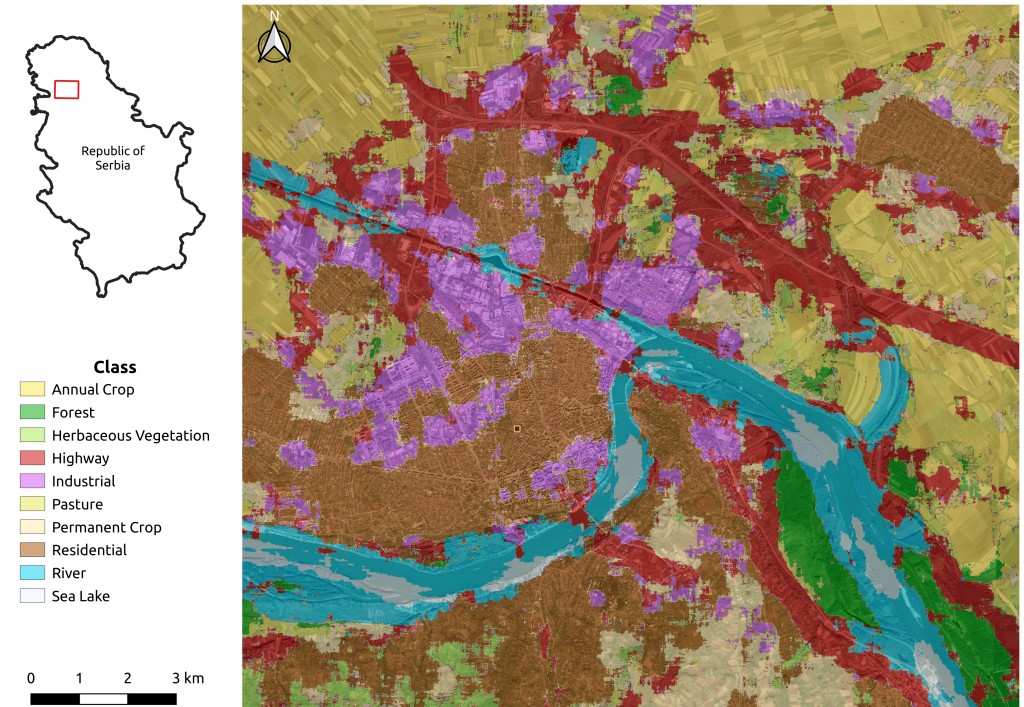

**Figure 4.** Sample land cover detection result.

### 2.3. Tracking Land Cover Changes

To generate images such as that shown in Figure 4, we apply a sliding window, with a single-pixel step, allowing us to infer the land-cover class for each pixel of a target image.

The full procedure used is the following:

- Obtain a multi-spectral (MS) image from Sentinel-2 for a desired rectangular polygon of any specific size;
- Pad the border of the MS image by mirroring 32 pixels along each border;
- Slide a 64 × 64 pixel window over the padded image, with a single-pixel step to generate the classification for all the pixels of the original image.

To monitor the changes in the land cover, one can perform the procedure for Sentinel-2 imagery taken at different points in time. To evaluate the applicability of the approach, we ran the analysis for a vast area stretching across Serbia (the area along the planned CER route), shown in Figure 5. We used images taken in six-month intervals from 2017 to the present day. Using two Ascend 910 NPUs, the inference takes about two weeks to complete. However, this needs to only be done once for the whole region, and we can continue to detect land cover for the subsequent Sentinel-2 imagery as it becomes available.

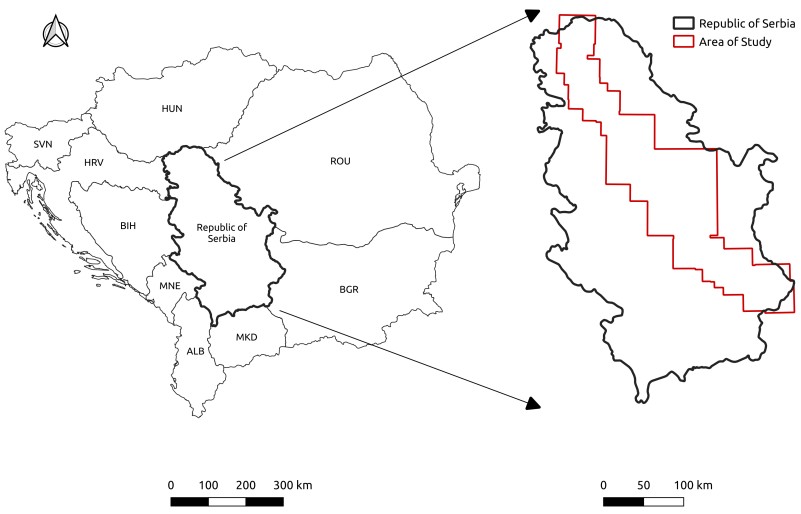

**Figure 5.** The geographical area of our study in Serbia.

## 3. Experiments and Results

To evaluate the applicability of the developed land-cover classification model as a tool to study the relationship between large transport infrastructure, land use, and the environment, we first need to establish a baseline. Therefore, we used it to analyze Sentinel-2 images obtained over a period of three years (2019 to 2021) for the region of Novi Sad, Serbia. The region is along the planned route of the CER express, but the first portion of the railway to be reconstructed as part of the development of this corridor has been opened in March 2022 (connecting Novi Sad with the Serbian capital—Belgrade). Thus, the region is, as of yet, not affected by the development of CER-related infrastructure and presents a good baseline for this and any study in the future.

As ground truth for our analysis, an LC map for the region was created manually by an experienced cartography expert. To produce an LC map with a very high level of detail, the expert combined visual information obtained for different bands. The resulting LC map is shown in Figure 6, together with the map obtained by AI, while the distribution of LC classes can be shown in Table 3.

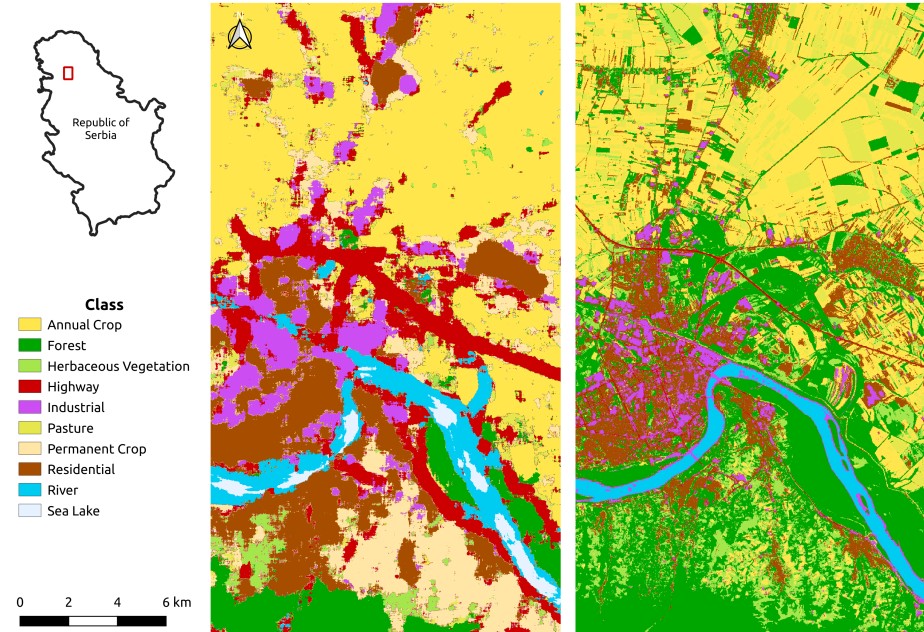

**Figure 6.** LC map created using the CNN vs. manual model.

**Table 3.** Distribution of land cover classes for ground truth vs. inferred.

| LC Classes Representation | | |
| --- | --- | --- |
| **Class** | **Ground Truth** | **Inferred** |
| Annual Crop | 38.1% | 34.0% |
| Forest | 24.0% | 7.7% |
| Herbaceous Vegetation | 4.6% | 7.8% |
| Highway | 0.05% | 13.7% |
| Industrial | 6.7% | 7.8% |
| Pasture | 13.1% | 4.4% |
| Permanent Crop | 0.0% | 6.5% |
| Residential | 11.1% | 11.8% |
| River | 1.6% | 6.2% |
| Sea-Lake | 0.0% | 0.5% |

The same satellite image that was used by the expert to generate the ground-truth map was then passed to the proposed automatic LC detection system. The output is shown in Figure 6, side by side with the GT map. As the visual inspection reveals, the ground truth is more detailed than the output of the model. This is likely due to the fact that the model used is trained on patches, rather than single pixel values. Clearly, the network trained on patches overestimates (dilates) classes such as the highway and river, corresponding to objects that are thin lines in the image, but seems to have reasonable accuracy when it comes to classes that cover larger areas of the image, such as residential, industrial and annual crop.

In fact, a per-pixel comparison of the two maps shows that the accuracy achieved is just 42%. A significant drop from the 97% patch classification accuracy, so a more in-depth analysis is needed.

The performance of the model, i.e., the confusion matrix, is shown in Figure 7. As we are interested in drawing statistics for the whole scene, more than per-pixel accuracy, Table 3 also shows the distribution of different LC classes in the ground truth and predicted image.

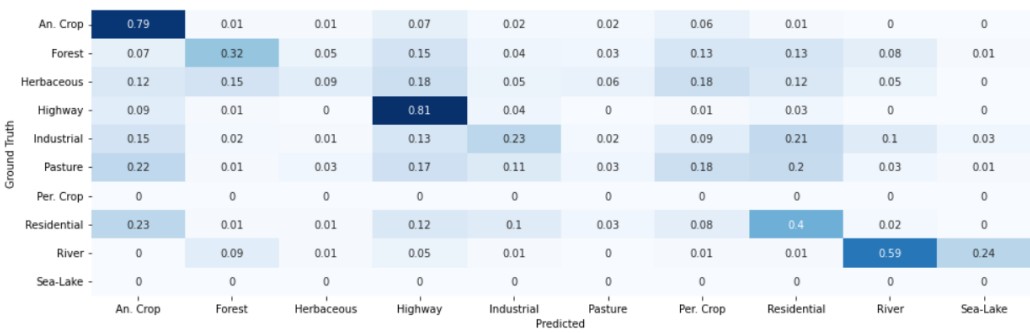

**Figure 7.** Performance of the segmentation model.

As our confusion matrix indicates, the classes predicted with the highest accuracy are *highway*, *annual crop* and *river*, with 81%, 79% and 59% accuracy. The *river* class is unsurprisingly mostly confused with the *Sea&Lake* class. In fact, when bundled together, these two classes could be detected with 83% accuracy. The errors for both the bundled *river + Sea&&Lake* class and the *highway* class tend to be fairly uniformly distributed across other classes, supporting the qualitative analysis conclusion that these structures are dilated in the inferred image.

The scene encompasses a large swath of agricultural land (mostly annual crops) which the model manages to identify well and the prevalence of the class makes sure that distortion due to the patch-based approach does not affect the accuracy so much. Distinguishing between the residential and industrial areas requires more precision, so there

is significant confusion between these two types of land cover. Finally, the mountainous region south of the city has very intricate detail and is mostly covered by a combination of permanent crops and forests, which the model could not separate with sufficient precision.

However, looking at the distribution of different land cover in Table 3, one can see that, for the primary classes of interest, when it comes to the evaluation of the effect of the transport infrastructure on land use in the region (i.e., *annual crop*, *residential* and *industrial*), the difference between the ground truth and predicted distribution is between 0.6% and 3.9%, which is tolerable. Therefore, while the model is not perfect, we argue that it can be used as a basis for further time-series analysis of land cover changes, by restricting the number of classes we are analyzing to these three.

### 3.1. Land Cover Change Analysis

To analyse the changes in land use in the region of interest, we run our model on images taken every month over a 3-year-long period (2019 to 2021).

Land cover changes exhibit certain trends over the 3-year period studied, mostly due to seasonal changes in vegetation. Figure 8 shows the plot of the changes in the distribution of the LC classes, for all classes, over a period of three years.

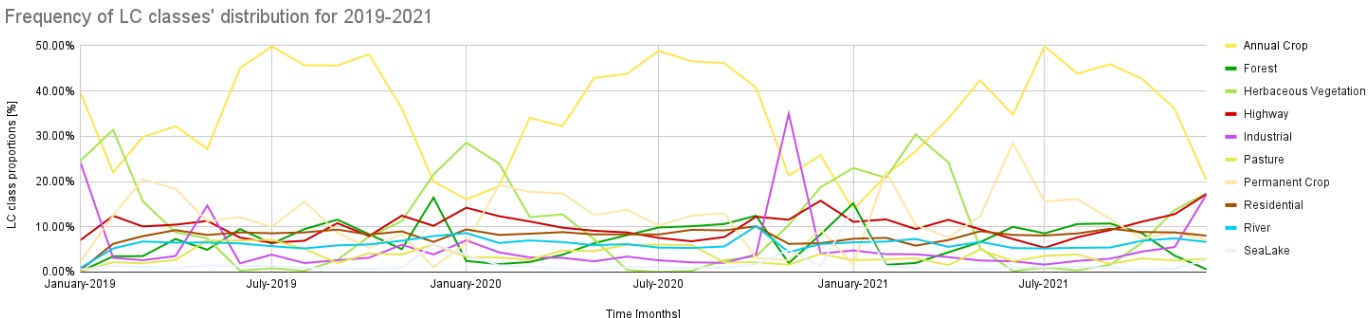

**Figure 8.** Timeline of LC frequencies for the indicator classes.

A similar graph restricted to the classes that the model predicts most reliably (residential, industrial and annual crop) is shown in Figure 9.

**Figure 9.** Distribution of indicator classes for 2019–2021.

In general, aside from a couple of obvious peaks, the percentage of the classes follows a steady periodical trend over the course of the year, for the entire three-year period. Upon

visual inspection, the occurring peaks are due to the noise in the data, such as cloud coverage and/or snow coverage, which the model does not handle. Therefore, the peaks occur mostly in winter. The most notable example of such a peak occurs in the month of November of 2021, where the entire polygon is clearly composed of two different satellite passages, one of which is very much contaminated by CC. The difference in the image quality is represented in Figure 10, which also contributes to the understanding of why there are unexpected peaks in the inference graph.

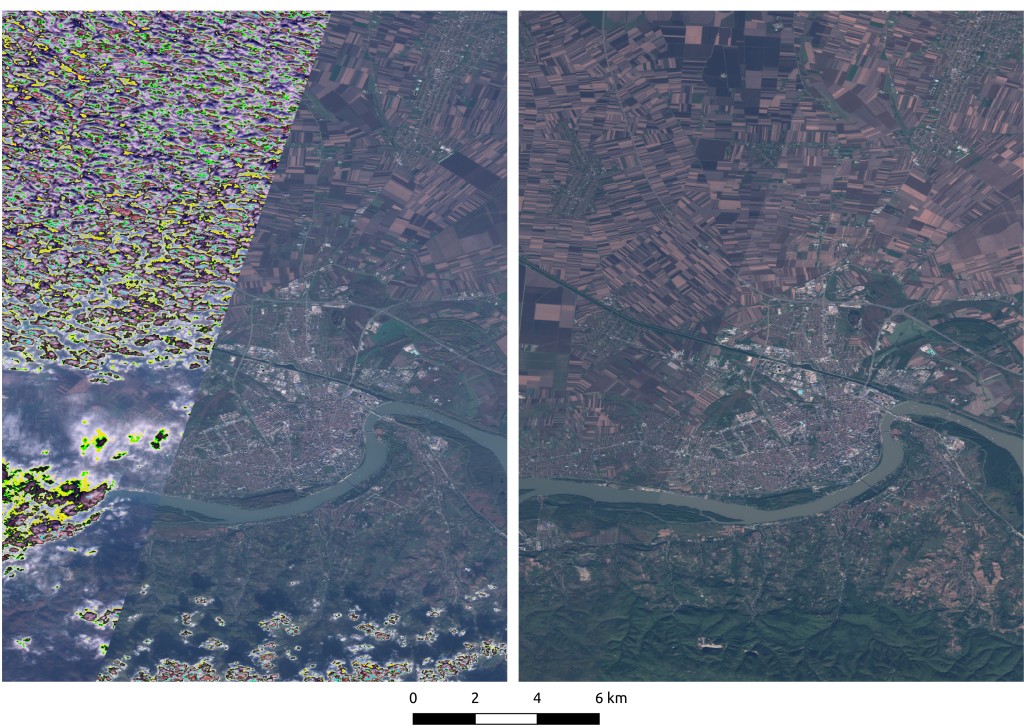

**Figure 10.** Novi Sad region for October/November of 2021—high vs. low CC.

However, we are interested mainly in the general trend of LC change, so the data have been aggregated on an annual level, by calculating the mean value of a given class, over the course of 12 months, for each year observed. Figure 11 shows a plot of the trends observed, while the mean annual values for the portion of the three classes of interest are shown in Table 4.

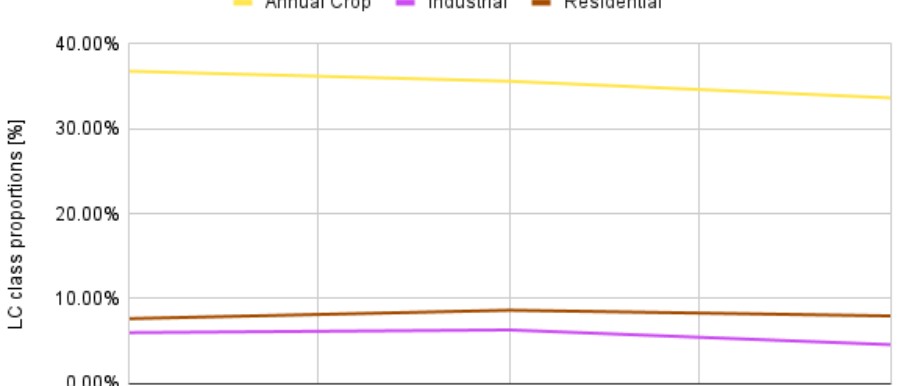

**Figure 11.** Annual changes of the relevant classes for 2019–2021.

**Table 4.** Trends of relevant LC classes for 2019–2021.

| Average Frequency of LC Classes per Year | | | |
|---|---|---|---|
| **Class** | **2019** | **2020** | **2021** |
| Annual Crop | 36.79% | 35.61% | 33.65% |
| Industrial | 5.97% | 6.29% | 4.56% |
| Residential | 7.62% | 8.61% | 7.94% |

As Figure 11 shows, the portion of annual crops is slightly decreasing over the course of the three years observed, which intuitively makes sense given that the examined polygon contains the second-largest city in the country of Serbia—Novi Sad. The slight reduction in the industrial category is also observable, while the residential holds fairly steady.

Our baseline, therefore, exhibits no large LC changes on an annual level in the period observed. This, however, may, in part, also be due to the effects of the COVID-19 pandemic.

### 3.2. Analysing LC Changes a Region in Central China

To further validate the applicability of the proposed approach, one should try to apply it to a region along the CER as geographically displaced from Novi Sad as possible. Thus, a region of interest (ROI) in central China was selected, for the final part of our study. The ROI lies along the CER railroad and is close to the city of Xiangyang, 300 km to the North-West of Wuhan. An example of a false-color image, overlaid with the inferred LC mask, is shown in Figure 12.

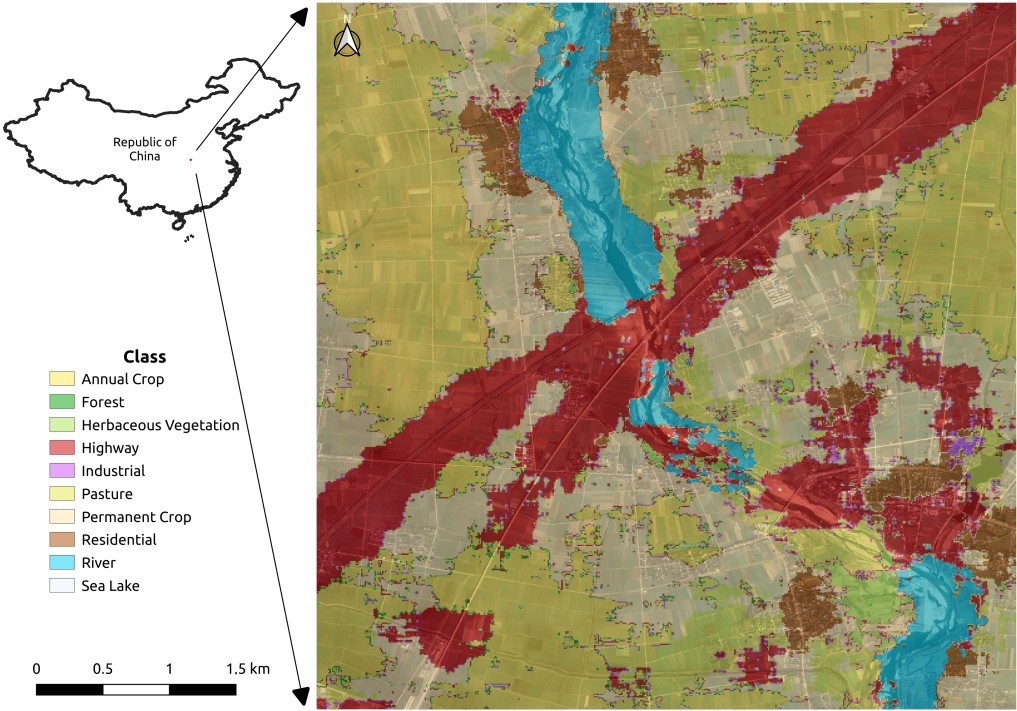

**Figure 12.** False-Color LC map of a region along the CER, located in Central China.

Following the methodology proposed in Section 3.1, we use our model to infer LC using Sentinel-2 images taken once a month, over the period of three years, from 2019–2021. As the aggregate indicator of land use, we, once again, use the distribution of different land cover classes. The graph of their changing distribution in the period observed is shown in Figure 13. As in the Novi Sad case, some peaks can be observed in the graph, which, upon manual inspection, originate from high cloud coverage, which the model was not designed to handle. We decided to eliminate such scenes from our analysis (this

can in fact be done automatically as the metadata attached to Sentinel-2 imagery have an indicator of the extent of cloud cover). For the months excluded, the class frequency data are interpolated between the neighboring months, so that a more smooth graph line can be obtained. After this correction, we extract the relevant classes and obtain a new graph and the corresponding trend, which are shown in Figures 14 and 15, respectively.

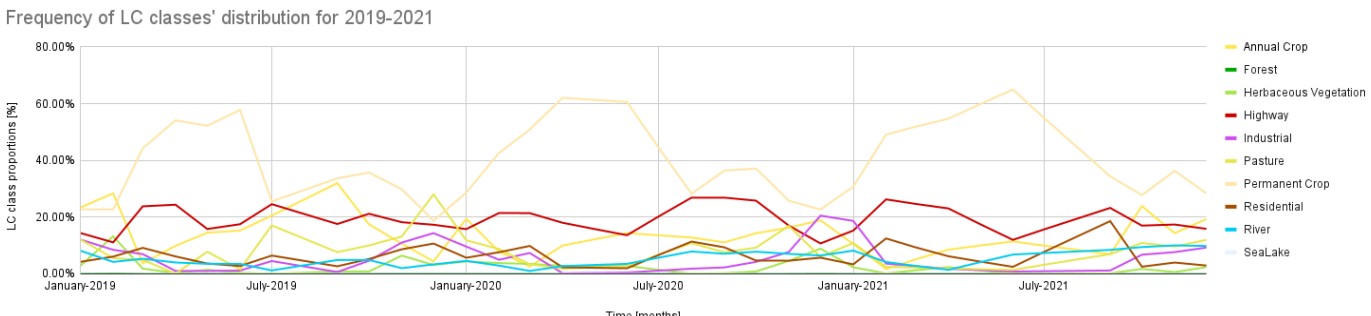

**Figure 13.** Dynamics of the distribution of LC classes for CER region in China, from 2019 to 2021.

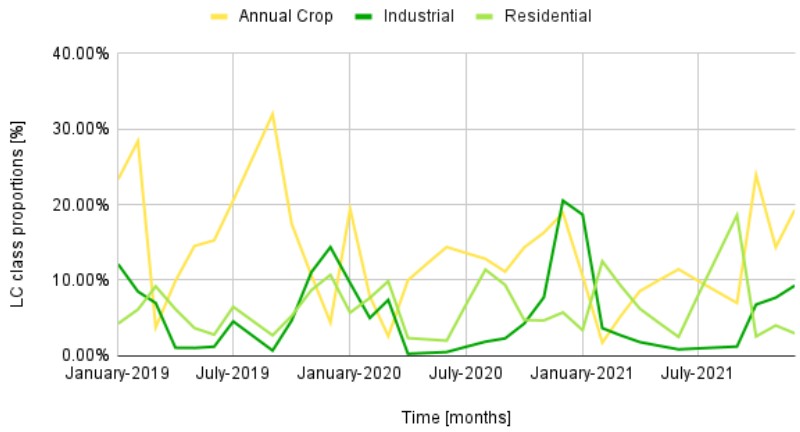

**Figure 14.** Corrected dynamics of the distribution of LC classes for the observed CER region in Central China, from 2019 to 2021.

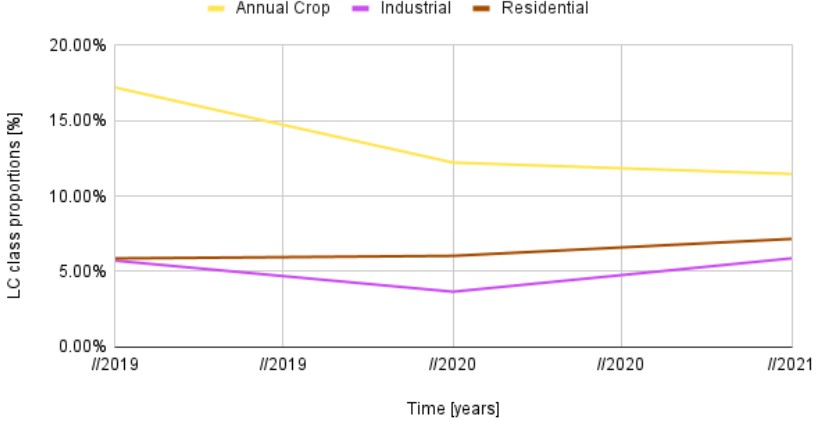

**Figure 15.** Annual aggregates of LC classes for the observed CER region in Central China, from 2019 to 2021.

The trends displayed in the graph indicate that there is a tendency for land use change, which can be summarized as the reduction of the agricultural activities, with a slight increase in residential and industrial development, in the observed region. The values plotted in Figure 15 are also listed in Table 5 to enable closer examination, supporting the same conclusion. Our model seems to indicate that, in the year 2020, there was a drop in both agricultural and industrial activity. Further analysis is needed to understand if this is just due to the error of the methodology, or whether the model relies on industrial-activity-related features to detect industrial regions, which were likely to be most severely affected by the COVID-19 pandemic in 2020.

**Table 5.** Trends of relevant LC classes for 2019–2021 for the CER region.

| Class | Average Yearly % for LC Classes | | |
|---|---|---|---|
| | 2019 | 2020 | 2021 |
| Annual Crop | 17.21% | 12.22% | 11.47% |
| Industrial | 5.71% | 3.65% | 5.87% |
| Residential | 5.86% | 6.03% | 7.16% |

## 4. Discussion

### 4.1. The Effect of Transport Infrastructure on Land Use

Kasraian et al. [2] provide a good overview of studies on the topic, relevant to our own research.

When it comes to railroads, several studies highlighted by the Kasraian et al. deal with the period stretching for over a hundred years, going back to the first development of railway in certain regions. Akgüngör et al. [29] and Beyzatlar and Kuştepeli [30] show that the development of railway in Turkey was strongly correlated to population growth along it and that, in the second half of the twentieth century, it had both short and long-term impacts on population density.

However, as Kasraian et al. observe the relationship between railway infrastructure development and population density, i.e., land use, can vary substantially across different territories in the same country. Da Silveira et al. [31] uncovered that more affluent regions in Portugal experienced population growth during the early development of the railways, but that it led to depopulation in poorer regions over the same period. On the other hand, Schwartz et al. reported a positive impact in the less developed parts of southwest France in the 1800s [32].

In a study that was conducted in Finland by Kotavaara et al. [33], it was concluded that the increase in population for period from 1920 to 1970 was profoundly related to the proximity to railways, besides during the 1930s, which was impacted by the global recession. In a different study, carried out by the same authors [34], it was expressed that the increase in population in the 1970s was heavily associated with the amount of time necessary to travel to the nearest railway station, but this factor was negligible for the population numbers regarding the last decades of the twentieth century. Furthermore, for the period from 2000 till 2007, the dependency between the increase in population and closeness to the railway station became significant again, which was explained by the large-scale investments from the government authorities in long distance transport infrastructure.

A more recent study of Chen et al. focuses specifically on the changes in land use in urban China, due to the development of High-Speed Rail (HSR) [3]. Their results, based on data for 238 cities in China, confirm that HSR development does play a significant role in facilitating the change in the structure of urban land use in China. They also show that the impact differs among various types of cities identified in the study. One of their conclusions is that, in order to improve urban land use efficiency, the land use policy related to HSR development needs to be implemented cautiously with a consideration of the specific conditions of different cities, i.e., that careful monitoring of the effects of

infrastructure development should be carried out to be able to adjust the policy as needed. Rather than using remote sensing in a conventional sense, the study used transaction data obtained from the website of the Land Market of China (www.landchina.com, 20 April 2022). While the authors claim that the dataset provides them with more accurate and consistent coverage of the land use volume change both spatially and temporally than remote sensing data's merit, it is hard to envision how the approach could be scaled to something with global coverage. Their result does suggest, however, a significant limitation of the results presented in our study, as we focus on only two areas along the CER express, so there is a significant chance of our sample to be too small to enable drawing of any general conclusions. We, therefore, do not attempt to do so but rather focus on providing a DL tool that can be used in broader studies and further improved, by both our group and other researchers.

As Kasraian et al. [2] observe, the indicator of land-cover change has been seldom been used in studies of relationships between transport infrastructure and land use, which adds significance in terms of impact to the study presented here. The results of the few studies Kasraian et al. [2] uncovered, which use the indicators of land cover, were quite contradictory. The distance to the railway line was reported to have no impact on land-cover change in Guangzhou from 1979 to 1992 [35]. In the case of Nanjing (1988–2000), the proximity to rail was even shown to discourage the conversion to urban land. The authors of that study attribute this to the fact that rail does not support within-city displacements but serves "long-distance interurban" commutes [36]. Again, to be able to resolve these inconsistencies between different studies, we will need to extend our analysis to a much larger portion of the CER express affected region.

Although the bulk of the studies support the claim that development of railways increases population density along them and causes urbanization of the land, it should be noted that railways, along with water transport, represent the cleanest ways of moving goods and people [37]. Thus, any negative effect on the environment in terms of urbanization and industrialization, as evidenced by the change in land cover and suggested by our own results to be occurring in the Xiangyang region, should be evaluated in terms of the benefits of using rail transport rather than some other modes that are more harmful to the environment.

### 4.2. Deep Learning for Remote Sensing

DL deals with the science and practical aspects of training and deploying neural networks with a large number of layers of artificial neurons and has been enabled by the advances in algorithms (for training such systems), hardware (Graphical Processing Units—GPUs), and large amounts of data that are generated online. The structure of the neural networks and the back-propagation algorithm used to train them makes them singularly well suited for parallel execution on GPUs, allowing the systems with a large number of neurons to be trained on very large amounts of data in a reasonable time frame. For example, the largest DL models in use today are the transformer architectures used in natural language processing, such as the recently proposed Pathways Language Model (PaLM) [38], which has 540 parameters (weights of the neural network) and has been trained on 780 billion tokens (words and programming language constructs). This, however, is extreme and the training of such a model required the use of 6144 Tensor Processing Units (TPUs), which are hardware solutions specifically designed for deep learning and produced by Google. A number of hardware producers now provide hardware for DL. In the present study, we rely on Huawei Ascend Neural Processing Units (NPUs) for the training and deployment of our models. This choice was due to the existence of an established infrastructure at the Institute for AI R&D of Serbia, which uses Huawei equipment at its premises, among others.

One of the first areas of AI to benefit from the power of deep neural networks is computer vision. In 2012, Alex Krizhevsky et al. [24] trained a deep model for image classification within the scope of the IMAGENET challenge, which resulted in an improvement of

30% over the state of the art at that time. The architecture, which is a Convolutional Neural Network (CNN), has become known as Alexnet and is still in use today, despite subsequent proliferation of models used in computer vision. The next IMAGENET challenge saw several deep-learning-based entries, including an architecture based on Alexnet, but with twice as many layers (16), proposed by the Visual Geometry Group at the University of Oxford, which had improved performance [22]. This architecture, aptly named VGG16, represents one of the most influential and widely used CNN architectures [23] and is the one that we use in the present study.

Traditional neural networks, such as the Multi-layer Perceptron (or as Yuan et al. dub it the Back-propagation Neural Network) and the Generalized Regression Neural Network, have been in use in the area of remote sensing for a long time and achieved notable success, but their applications were limited by the performance that could be achieved with these "shallow" models [15].

The main advantage of DL approaches, over the more classical, machine learning, computer vision and remote sensing methods, is in their end-to-end training. This means that they do not rely on human-engineered features to decide, but take the visual input in its "raw" form and are able to learn the features relevant to the task themselves [13]. In satellite-based remote sensing, the input takes the form of a multispectral image and the training is usually done based on data labeled by humans. Provided that enough such data are presented to the network, it is able to extract relevant low-level and high-level features and capture complex relations present in multidimensional remotely sensed images [39,40].

When it comes to the processing of visual data, there is a number of "mainstream" deep neural network architectures which are commonly used and they include autoencoders [41], Generative Adversarial Networks [42], Convolutional Neural Networks (CNNs) and visual transformers [43]. The first two approaches require no supervision from the humans but have the drawback of lower performance in general. The last is a fairly recent development inspired by architectures typically used for natural language processing. Of the four, CNNs are the most used deep learning algorithm for spatial pattern analysis. They are designed to detect specific spatial features in images, such as corners, borders, texture, edges and more abstract shapes, learning the parameters of a bank of filters which are applied to the image (in the first convolutional layer) or the results of processing of the previous layers using convolution—hence their name.

A CNN represents a network of neurons that are organized in a number of layers. In a remote sensing application, such as the one presented in this study, the first is the input layer, i.e., the layer that receives remote sensing Earth observation data, and the final layer is the output layer, which e.g., produces a label for a pixel, corresponding to the class of land cover that is observed in that location. Between the input and output, there is a number of hidden layers (convolutional, pooling and fully connected layers) which project the input into the feature space which allows the fully connected layers to learn to classify it. Although the number of parameters in a typical CNN is large (138 million for the VGG16 architecture we used), the design of the CNN actually significantly reduces the number of parameters when compared to the "naive" approach of creating a network with the same number of fully connected layers. Even so, the amount of data required to train a CNN is significant, as Zhang et al. experienced. In the study, they applied a combination of an MLP and a fairly shallow CNN (just 4 layers), as they did not have enough data to train a deeper model [44].

In deep learning, the problem is usually circumvented using transfer learning, where a smaller data set is used to "fine-tune" a model initially trained on a different, much larger data set. It is common for every deep learning framework to have a store of such models available, sometimes referred to as a model zoo (e.g., https://pytorch.org/serve/model_zoo.html, 15 April 2022).

However, transfer learning based on models available in the computer vision community is of little use in remote sensing, as the data in Earth Observation (EO) have different properties. The position of the sensors is different, i.e., the overhead perspective in EO data

is very different from the side view perspective usual in "natural" images. Natural images are typically three channel RGB images, while, in EO data, they are often a multi-channel image. In natural images, objects tend to be in the centre of the image and in high resolution, whereas, in EO data, the objects are not centered and in low resolution. Finally, in EO data, objects or classes tend to be more densely packed and heterogeneous than in natural images [45].

Bearing this in mind, the development and sharing of pretrained models is vital to the advancement of the state-of-the-art of DL applications in the domain of remote sensing. The models, naturally, should be accompanied by papers outlining their performance and limitations in a specific application domain to make the whole process as efficient and effective as possible.

### 4.3. Limitations of the Research

While the study presented here validates the hypothesis that there is significant potential in applying state-of-the-art DL approaches to land cover classification as a remote sensing tool to automatically monitor the effects of large scale railway infrastructure on the land use and environment, it also shows that there are significant limitations to using them.

Most notably, the high land cover classification accuracy of the models trained and tested on the benchmark data sets such as the Eurosat, which rely on image patches, leads to a significant drop in performance, when tested in a scenario where each pixel needs to be classified. A qualitative examination of the labeling, however, shows that the effect of using such models is the dilation of structures with dimensions small relative to the patch size, but the more compact land cover classes can be fairly reliably detected. If the goal is to monitor the relative portion of such classes as industrial infrastructure, residential urban areas and stretches of greenery, the existing approaches still form a viable alternative. More precise land cover segmentation models are likely to succeed with such approaches as the DL methodology of choice when it comes to land cover change and environmental impact monitoring. The main limitation for the development of such approaches is that they currently require tedious manual labeling of land-cover for each pixel of numerous remotely-sensed images.

In terms of the environmental impact results presented in this study, the main limitation is the fact that this initial investigation has been carried out for just two areas along the planned and existing CER route. In that sense, the study is too limited to draw any general conclusions about the effect of CER on the environment, but this was not our goal. We simply wanted to develop a tool based on the state-of-the-art approaches that would serve as a basis for further studies and evaluate its usefulness in this important domain.

## 5. Conclusions

Land cover and land-use change monitoring are highly relevant in the context of managing resources and infrastructure and evaluating the impact of transport infrastructure on the environment and human activity. Having a usable and precise method to monitor these changes on a global scale is therefore of significant interest, both in the research community and from the standpoint of policymakers and land administrators.

Applications of Deep Learning to the area of remote sensing promise to usher in the are of new automatic tools that could help solve that problem. Significant barriers to the use of such tools for EO researchers and practitioners exist.

In the study presented here, we evaluate the applicability of a well-known convolutional neural network model in the context of monitoring the LC changes that are caused by the development of large transport infrastructure, such as the China-Express Railway (CER), which stretches across Eurasia, the largest continent on Earth. The proposed approach is based on the Sentinel-2 satellite imagery, which is publicly available for research.

To automatically analyze Sentinel-2 imagery in terms of land cover, we adapt standard deep neural network architecture (VGG16), trained on the Eurosat data set, containing 27,000 patches of the multi-spectral Sentinel-2 satellite images, labeled with 10 classes of

land cover. This yields a model that can predict the label of a patch with the accuracy of 97% when evaluated on a test set comprised of 30% of the Eurosat data.

To evaluate if such an approach can be used effectively used for LC monitoring, we then employ this architecture to create automatic LC maps for two regions along the planned CER route, a continent apart. Our analysis showed that, when compared to manually-generated ground truth, due to the patch-based training approach, the model is not able to match the level of detail of the manual annotation and the resolution of Sentinel-3 data, but can quickly and efficiently provide an indicator of the distribution of three land-cover classes of special interest in our study (annual crops, residential and industrial), as well as identify structures such as highways and rivers and lakes accurately, although it tends to make them larger than in reality.

Once the LC model was developed, we used it to process an image per month for each of the two regions of interest, over a 3-year period and try to infer the trends of LC change.

Our study shows that, while there is significant potential in applying DL for efficient monitoring and studies of our planet, there is significant work to be done in both the remote sensing and deep learning community. The computer vision models need to be adapted to handle the diverse, usually multispectral data, used in the remote sensing community, but their applicability also needs to be validated in a practical application scenario, as was done in the study presented here.

When it comes to the further research needs identified in the course of our study, it seems clear that to be able to fully utilize the resolution of Sentinel-2 imagery, one will need to explore the application of deep learning approaches designed specifically for segmentation, such as U-net [46]. As the first step, and bearing in mind that our results are significantly affected by the presence of clouds and their shadows in the images, we will attempt to apply the recent results of Jiao et al. [46] to remove tainted parts of the image.

**Author Contributions:** Conceptualization, D.C. and M.P.; methodology, S.I.; software, S.I.; validation, N.A.; formal analysis, N.A. and M.P.; investigation, M.P. and N.A.; resources, D.C.; data curation, S.I.; writing—original draft preparation, M.P.; writing—review and editing, M.P., D.C., N.A. and S.I.; visualization, N.A.; supervision, D.C.; project administration, S.I.; funding acquisition, D.C. All authors have read and agreed to the published version of the manuscript.

**Funding:** This work has in part been funded by the EU H2020 Programme (Grant #730381). Responsibility for the content of this publication lies with the authors.

**Data Availability Statement:** The data that support the findings of this study are openly available online, as part of the EuroSAT dataset, at https://github.com/phelber/EuroSAT, 19 April 2022.

**Conflicts of Interest:** The authors declare no conflict of interest.

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
