# Peer review of "Monitoring the Impact of Large Transport Infrastructure on Land Use and Environment Using Deep Learning and Satellite Imagery"

_remotesensing, doi:10.3390/rs14102494_

Round 1

Reviewer 1 Report

In the manuscript titled ", Monitoring the Impact of Large Transport Infrastructure on and Use and Environment Using Deep Learning and Satellite Imagery," the Authors employ a state-of-the-art deep-learning-based approach to automatically detect different types of land cover based on multispectral Sentinel-2 imagery. I believe that the methods used in the work are appropriate and well-chosen for the topic and clearly described. The work presents an appropriate structure, and the contents do not repeat. This topic fits the scope of the journal. I propose to reconstruct this document gently before publication.

  • No doubt, the authors presented a clear literature review in the introduction section. Still, I am afraid a five pages introduction section would make the readers bored and make this research uninteresting. It's better to modify the introduction and keep it in line with your topic by discussing the significance of the study, previous studies, and their methods, then what your research is making new or which research gaps/limitations the authors have covered with this research. Rest of the literature from here, the authors can move to the discussion section to support their results and topic.
  • Currently, authors presented their results without any reference, etc.; it's strongly recommended to add Section 4. Discussion and present and validate your results with previously done studies. Then also, add subsection 4.1 and describe the limitations of your research.
  • After adding the discussion section, conclude your research accordingly.
  • I found all the figures are presented ruffly; Please add the legends, scale, north arrow, etc., correctly.
  • Line 147. The authors mentioned, "We rely on HUAWEI Ascend Neural Processing Units (NPUs) for the training and deployment of our models.". I think it would be better if the authors added a few more words to describe the reason for choosing NPUs instead of other units.
  • Line 381. "Geometric correction." Please specify what kind of geometric correction the authors have done and why it was necessary, which will improve the readers' understanding.
  • How about the legends of Figures 6 and 7, and so on? Instead of a raw presentation, please add a Legend, scale bar, north arrow, etc.
  • It would be better to discuss section 1.3 in the methodology section. In the introduction section, the authors can highlight some significant points.
  • The quality of figure 5 is too low, and the legends are unreadable. Please consider modifying it.

  • Line 70-105 are completely plagiarised. Please consider rephrasing.
  • Line 42, Please update it with the latest launches of the GF series.
  • Line 88. Add a space between 2 sentences.
  • Please mention the source of figure 2.
  • Line 322. The resulting code can be accessed on GitHub?? 
  • Line 370. "He" region or "the region"
  • Line 399. [?] ???
  • Line 420. Figure ?? ???
  • Section 5. References. DOIs of all citations are missing. Please add the DOI for all the citations.

Reviewer 2 Report

Dear Authors and Editor,

I had the pleasure to read the article “ Monitoring the Impact of Large Transport Infrastructure on Land Use and Environment Using Deep Learning and Satellite Imagery”. The title and idea of this text were interesting for me because I really excepted there will be some explanation of this important path of railway lines on the earth. Step by step it turned out the authors focused half of their article on the explain what is deep learning – this is completely not needed and has no connections with the conclusions. Of course, some small definitions and explanations are useful but this text is definitely too long. In the paper, there is the figure of the CER lines but I really do not know what part of this map was covered by the analyses which were conducted in this research. There is no map of the area of interest for this region. Taking into consideration the results presented in figures 6-8 it looks like the accuracy of the used algorithm is generally weak. This can be clearly confirmed by figure 9 because it cannot be reliable that the highway presented in figure 9 sometimes grows and sometimes decrease. There is instant growth in the infrastructure of the world and this is obvious. By the way, on the graph, there is a missing description of Y-axis. Publishing the article in a high IF journal should not touch the obvious matter like high CC or low CC and presented it on the whole page – this is an obvious matter which does not need so big explanations.

The weaker point touches the chapter 3.2. Actually, the word “transport” doesn’t appear here at all. As far as I remember you were analysing the influence of transport on the land cover changes. Did you prove that in your results? What we can see is that the industrial and residential areas are growing a little bit during the analysed time interval however there are no explanations it is covered by transport. You’ve also omitted the changes in the transportation network itself.

The conclusions are very general which actually inform the reader that LC is important, DL is useful and the Sentinel data were effectively used. I’m not sure this is what you really want to present.

Returning back to the main idea of the article I definitely think it is interesting. I also appreciate the amount of work made by the authors however I think it is organized and presented in a weak way.

I wish authors further succeed in publication. Some small comments are presented below.

1-12

Some results should appear in the abstract.

13

I’m not sure to use VGG16 as a keyword.

129-247

The literature review is generally interesting but I’m not sure if there is a strong need to write so long chapters according to widely known deep learning ideas. I would suggest shortening it to this part which is strictly connected with the proposed methodology.

261-273

This is not needed.

280

As you mentioned in line 251 the resolution 10 metered of Sentinel data was used so what is table 1? How is it connected with the methodology?

346

I’m not sure about the definition of “land use land cover”. Land use or land cover? What is the difference? This is the first time I see such a name. Land cover is a very clear definition. What's more – you are using the definition of land cover in line 409, not LULC

366

This figure is really bad. Where is the legend? Where is the scale bar, and location map? What this square stands for? Why the resolution is so bad? The area of interest map should be one of the most important in the article – please make it better.

379

Why are you writing  what table 3 shows if it is written in the table 3 description in line 415

392

The figure 8 name should be more informative.

431

Please do not use in your paper such sentences as “the figure displays, the figure shows” These are sentences that are just below the figure so you do not have to repeat them in the text.

432

You should put CC abbreviation for cloud contamination

445

There is no evidence it is because of COVID-19

Round 2

Reviewer 1 Report

The authors have addressed all my suggestions, and the manuscript is now significantly improved and acceptable for publication.

Reviewer 2 Report

The authors responded positively to my comments and improved the whole text.
I wish the authors success in further publishing.